# B4GALT1 as a New Biomarker of Idiopathic Pulmonary Fibrosis

**DOI:** 10.3390/ijms232315040

**Published:** 2022-11-30

**Authors:** Claudia De Vitis, Michela D’Ascanio, Andrea Sacconi, Dario Pizzirusso, Valentina Salvati, Massimiliano Mancini, Giorgia Scafetta, Roberto Cirombella, Francesca Ascenzi, Sara Bruschini, Antonella Esposito, Silvia Castelli, Claudia Salvucci, Leonardo Teodonio, Bruno Sposato, Angela Catizone, Arianna Di Napoli, Andrea Vecchione, Gennaro Ciliberto, Salvatore Sciacchitano, Alberto Ricci, Rita Mancini

**Affiliations:** 1Department of Clinical and Molecular Medicine, Sant’Andrea Hospital, University of Rome “Sapienza”, 00185 Rome, Italy; 2UOC Respiratory Disease, Sant’Andrea Hospital, 00189 Rome, Italy; 3UOSD Clinical Trial Center, Biostatistics and Bioinformatics, IRCCS Regina Elena National Cancer Institute, Via Elio Chianesi 53, 00144 Rome, Italy; 4Scientific Direction, IRCCS Regina Elena National Cancer Institute, Via Elio Chianesi 53, 00144 Rome, Italy; 5Morphologic and Molecular Pathology Unit, S. Andrea University Hospital, 00189 Rome, Italy; 6Department of Experimental and Clinical Medicine, Magna Graecia University of Catanzaro, 88100 Catanzaro, Italy; 7Division of Thoracic Surgery, Sant’Andrea Hospital, University of Rome “Sapienza”, 00185 Rome, Italy; 8Pneumology Department, Azienda USL Toscana Sud-Est, “Misericordia” Hospital, 58100 Grosseto, Italy; 9Department of Anatomy, Histology, Forensic-Medicine and Orthopedics, Sapienza University of Rome, 00161 Rome, Italy

**Keywords:** B4GALT1, idiopathic fibrosis pulmonary, lung cancer, EMT, lung fibrosis, immunohistochemistry, mRNA

## Abstract

Idiopathic pulmonary fibrosis (IPF) is a disease characterized by progressive scarring of the lung that involves the pulmonary interstitium. The disease may rapidly progress, leading to respiratory failure, and the long-term survival is poor. There are no accurate biomarkers available so far. Our aim was to evaluate the expression of the B4GALT1 in patients with IPF. Analysis of B4GALT1 gene expression was performed in silico on two gene sets, retrieved from the Gene Expression Omnibus database. Expression of B4GALT1 was then evaluated, both at the mRNA and protein levels, on lung specimens obtained from lung biopsies of 4 IPF patients, on one IPF-derived human primary cell and on 11 cases of IPF associated with cancer. In silico re-analysis demonstrated that the B4GALT1 gene was overexpressed in patients and human cell cultures with IPF (*p* = 0.03). Network analysis demonstrated that B4GALT1 upregulation was correlated with genes belonging to the EMT pathway (*p* = 0.01). The overexpression of B4GALT1 was observed, both at mRNA and protein levels, in lung biopsies of our four IPF patients and in the IPF-derived human primary cell, in other fibrotic non-lung tissues, and in IPF associated with cancer. In conclusion, our results indicate that B4GALT1 is overexpressed in IPF and could represent a novel marker of this disease.

## 1. Introduction

Idiopathic pulmonary fibrosis (IPF), which primarily affects older male smokers, is the paradigm of chronic progressive idiopathic fibrosis. It is characterized with a histopathological and radiological point of view by the usual interstitial pneumonia (UIP) pattern [1,2]. IPF’s natural course is unpredictable and the pathophysiological mechanisms underlying the disease, the anatomical remodeling and progression, are currently under investigation [3]. More relevant, accurate biomarkers to prognosticate and guide management for patients affected by this disease are still needed. Current paradigms into IPF pathogenesis, a disease as lethal as malignant disorders, hypothesize altered apoptosis mechanisms as well as changes in cellular and intracellular pathways [1,2]. Epithelial damage, fibroblasts activation and proliferation, cell senescence, and aberrant immunological responses were all considered as relevant in this complex scenario [2,4]. Activated fibroblasts, the myofibroblasts accumulated into fibroblastic foci, are the histologic hallmark of the disease and are the leading edge of fibrosis. The deposition of the extracellular matrix (ECM) by myofibroblasts is essential for fibrotic lung remodeling in IPF. The process leading to the transformation of fibroblasts and maintenance of their uncontrolled activity remains largely unknown [5,6]. The expansion of myofibroblasts within lung interstitial space may be due to exaggerated resident fibroblast proliferation, homing of circulating fibrocytes into the lung, and/or activated type II alveolar cells that migrate into the interstitial space assuming a mesenchymal phenotype, a phenomenon known as epithelial to mesenchymal transition (EMT) [7,8]. EMT is a dynamic, reversible phenomenon, active during embryogenesis but implicated in wound repair cancer metastasis and fibrosis [3,4]. Several pleiotropic signaling factors can initiate EMT, promoting repression of epithelial features and activating different mesenchymal genes responsible for cell-to-cell adhesion and migration. Transcription signaling factors, such as SNAIL, ZEB, and Twist are associated with the morphological changes of EMT [9].

The B4-galactosyltransferase (B4GALT) is a family composed by seven members with specific and distinct tissue distribution, function, and chronological expression. B4GALTs compete with each other to promote the expression of biologically active carbo-hydrate determinants by glycosylated glycans [10,11]. The B4-galactosylation of the glycans is important in many physiological events, but it is also observed in the development, invasiveness, and drug resistance of different cancer cell types [12,13,14]. Among B4GALTs the B4GALT1 is an emerging gene involved in cancer stem cell propagation [5]. Furthermore, B4GALT1 led to the formation of the sialyl-Lewis X determinant, which was used as a tumor marker for lung cancer, but is also expressed by pulmonary epithelial cell in patients with IPF [15,16]. In consideration of the complex role of the B4GALT family in the control of cell function, we analyzed the expression of B4GALT1 in primary fibroblast cell culture and in tissue specimens from IPF lungs [17].

## 2. Results

### 2.1. B4GALT1 Gene Expression Is Increased in IPF and It Correlates with EMT Gene Pathway

The gene expression of B4GALT1 was studied in IPF tissue samples obtained from human lung biopsy and the patient’s respiratory epithelial cell lines, previously published [5], retrieved by two different gene sets from the Gene Expression Omnibus (GEO) database. B4GALT1 was statistically overexpressed in patients with IPF, and to a lesser extend, in chronic hypersensitivity pneumonitis (CHP), as compared to healthy controls (Figure 1a). The gene expression of B4GALT1 was even more evident when the analysis was performed on another gene set, obtained from human epithelial lung cell cultures, derived from patients affected by IPF and compared to normal human lungs (Figure 1b). Further analysis indicated that B4GALT1 gene expression was correlated with genes belonging to the EMT pathway (Figure 1c, Appendix A). The analysis of the network involved in the B4GALT1-associated EMT pathway indicates the involvement of several genes, as reported in Table 1, where the specific degree of significance is indicated for each on of them. Among them, the strongest associated pathways are those related to focal adhesion (WP306), miRNA targets in ECM and membrane receptors (WP2911), PI3K-Akt signaling pathways (WP3932, WP4172), as well as those involved in lung cancer (Senescence and Autophagy, WP615; cancer and small cell lung cancer, WP4658) and lung fibrosis (WP3624). Graphic network analysis illustrates the correlation of B4GALT1 expression with 55 genes strongly upregulated, belonging to the EMT pathway (Figure 1d).

All together, the results of such in silico analysis indicated that B4GALT1 RNAs’ expression is upregulated in both tissue samples and cell cultures from patients with IPF and it is associated with EMT gene network. These results suggest a possible role of this gene as a biomarker of lung fibrosis.

### 2.2. Protein and Gene Expression Analysis in Primary Fibroblast Cell Cultures from IPF Patient

IPF-derived human fibroblasts primary cells, obtained from one patient admitted to our university hospital, were isolated and grown in appropriate medium culture. These primary cells displayed several morphological features that differed from those observed in normal ones. They appeared more flattened and jagged than the control fibroblasts (Figure 2a), with a higher growth rate.

In addition, immunofluorescence analysis revaled a different and irregular distribution of intracellular actin fibers, as compared to the parallel regular distribution observed in normal fibroblasts (Figure 2b). To confirm the immunophenotype pattern of IPF, two specific markers where evaluated by immunohistochemistry (IHC). Both of them, namely α-SMA and CK7, where highly expressed (Appendix A). B4GALT1 gene overexpression, observed by in silico analysis in IPF samples, was confirmed by the analysis performed in the lung biopsy obtained from the same IPF patient donor of the primary cell line (Figure 2d). The reaction was clearly evident in the border area between fibrosis and normal alveolar spaces. Activated pneumocytes, fibroblasts, and neo-formed vessel figures were clearly positive at B4GALT1 immuno-staining (Figure 2c). RNA expression analysis, performed on fibroblasts grown in IPF primary culture, revealed upregulation of NANOG, OCT4 e CD133, ß-catenin and SLUG genes, and downregulation of SNAIL, compared to control normal fibroblast cell culture (Figure 2e). These preliminary data confirmed the upregulation observed by in silico re-analysis and indicated that B4GALT1 is overexpressed both at the mRNA and protein levels in the lung on our IPF patient.

### 2.3. Protein Expression Analysis in Lung Speciemens from IPFpatients

We then extended our analysis regarding the expression of B4GALT1 protein to a group of 4 patients affected by IPF (Table 2). B4GALT1 immunoperoxidase staining in IPF lung tissue showed moderate to strong expression both in pneumocytes from areas of active remodeling and in fibroblast embedded in the sclerotic areas. Endothelial cells and inflammatory cells showed a much lower degree of staining intensity and unaffected lung tissue was negative (Figure 3a–c).

### 2.4. Protein and Gene Expression Analysis in Patients with Fibrosis in Other Non-Lung Tissues

We then asked whether the increased expression of B4GALT1, observed in lung tissues affected by fibrosis, was a finding specific for the lung. For this reason, we analyzed three other cases of each human pathological tissue with fibrosis-related changes, such as thyroid, breast, skin, and liver. We observed an increased expression of B4GALT1 in epithelial cells in fibrotic areas of all these tissues, with a special high expression level in the cirrhotic liver (Appendix A). The increased expression of B4GALT1 in all these organs with fibrous changes/remodeling suggests that the upregulation of B4GALT1 is a common responsive mechanism to fibrous remodeling in all these organs.

### 2.5. B4GALT1 Is Overexpressed in UIP/IPF Associated with Lung Cancer

Since some studies suggested a possible association between UIP/IPF and lung cancer, we asked whether B4GALT1 expression was also expressed in human lung tissues from a patient with both UIP/IPF and lung cancer. To this purpose, we selected specific areas where fibrosis and neoplastic trasformation were detected in association and we checked for B4GALT1 protein expression. B4GALT1 showed strong immunoperoxidase stain in the cytoplasm of neoplastic lung cells, in line with previously published data from our group [1] (Figure 2d). Interestingly, B4GALT1 was also overexpressed, albeith at a lesser level, in reactive pneumocytes located in fibrous remodeled lung parenchyma nearby the tumor (Figure 2e).

## 3. Discussion

IPF is a disease generally characterized by a poor prognosis and such a condition is often responsible for a progressive loss of lung function with a fatal event occurring approximately 3 years from diagnosis in untreated patients. Its behavior is similar to that reported for lung cancer [3]. According to recent evidence, patients affected by IPF have a nearly five-fold increased risk of developing lung cancer, as compared to the general population. In addition, the incidence of lung cancer in IPF patients increases over time, being 3.3% 1 year after diagnosis and racing to 54.7% after 10 years. In a previous study, we demonstrated that the expression of B4GALT1 was linked to tumor development and progression [5,8]. Lung cancer was prevalent and exerted a dramatic survival impact on patients with IPF [8]. However, the expression of B4GALT1 in IPF tissue and cell lines, as well as in UIP/IPF associated with lung cancer, was never tested before.

B4GALT1 is emerging as a new promising marker in different types of tumors, including bladder cancer, where it could be used as a predictive marker for the choice of ACT in pT3/4 or N+ patients [9], in colorectal cancer (CRC),where its overexpression is a potential independent adverse prognostic factor for the overall survival [10] and where its aberrant expression and methylation status can be used as a prognostic and therapy response predictive indicator [11], and in testicular cancer, where its expression in peripheral T-lymphocytes may be useful as a marker of primary progressive or relapsed disease [12]. Moreover, B4GALT1 knockdown was responsible for an increase in apoptosis and autophagy of glioblastoma both in vitro and in vivo [13]. According to this study, BAX, BCL-2, cleaved caspase-3, Beclin-1, and LC3 may be downstream target effectors of B4GALT1 involved in apoptosis and autophagy. B4GALT1 was proposed as a new therapeutic biomarker in this type of tumor. Finally, experiments performed in an orthotopic and heterotopic pancreatic ductal adenocarcinoma (PDAC) model, demonstrated that B4GALT1 belongs to a new signaling pathway, including p65 upstream and CDK11p110 downstream, respectively, involved in chemoresistance of pancreatic cancer [14].

Our study demonstrated that B4GALT1 is overexpressed at mRNA and protein levels in lung specimens from IPF patients. Its expression is not limited to epithelial cells, but can be detected also in immune cells and endothelial cells. These results suggest a possible role of B4GALT1 in the EMT process, which is known to be implicated in both pathogenesis fibrosis and cancer. This hypothesis is confirmed by our in silico analysis that revealed a strong connection of B4GALT1 gene expression with the EMT gene network. In addition, our data obtained in the IPF-derived primary cell line indicate that the deregulation of two different sets of genes is associated with B4GALT1 expression; in particular, those linked to staminal condition (NANOG, OCT4, and CD133) and those belonging to the EMT process (ß-catenin, SNAIL, and SLUG). All these results point toward the role of B4GALT1 in the basic pathological process in common with fibrosis and cancer and suggest the key role of this gene in both conditions.

The role of B4GALT1 in cancer is not surprising and is not new. B4GALT1 belongs to the beta-1,4-galactosyltransferase (beta4GalT) family of genes and it catalyzes the transfer of UDP-galactose to acceptor carbohydrates [15,16]. Many different observations indicate that dysregulation of glycan-related genes, responsible for aberrant expression of glycan structures, is associated with cancer development and progression [17,18,19,20,21].

In this regard, there are close similarities between the LGASL-3 gene, encoding galectin-3 (a member of the galectin family of carbohydrate binding proteins), and B4GALT1. Galectin-3, also known as galactose-specific lectin 3, specifically binds β-galactosides, including galactose, lactose, poly-lactosamine, and N-acetyl-lactosamine (LacNAc) [22]. B4GALT1 transfers galactose from uridine diphosphate-alpha-D-galactose (UDP-galactose) to acceptor sugars, such as N-acetyl glucosamine (GlcNAc) [23]. Galactoses and galectins are both involved in inflammatory processes and cancer, and it was observed that without galactose, the present-day galectin a priori would never develop [24]. Galectin-3 is involved in many different diseases and conditions, including cancer and fibrosis [25,26]. Moreover, a roles of galectin-3 in IPF was demonstrated [27,28], and inhibitors of galectin-3 were recently proposed to treat fibrotic disease as well as IPF [29,30]. Both galectin-3 and B4GALT1 may affect signaling pathways involved in cell–cell communication, in tumor cell dissociation and invasion, in cell-matrix interactions, in tumor angiogenesis, in immune modulation, and in metastasis formation. Our current study indicates that B4GALT1 is involved also in fibrosis, as previously reported for LGASL-3. B4GALT1 could indicate a novel potential therapeutic target in fibrosis as well.

## 4. Materials and Methods

### 4.1. Study Population

Fifteen patients with IPF from Sant’Andrea University Hospital between May 2018 and September 2021 were enrolled. For all patients, we collected data of the main epidemiological characteristics, smoking habit (measured in p/Y), histological diagnosis, respiratory function tests, as well as radiological investigations at the time of biopsy. Patients were divided into the UIP and UIP + lung cancer (LC) groups. In both groups, the diagnostic process included CT-scan and CT-guided lung biopsy or surgical video thoracoscopy (VATS) to obtain tissue specimens for the analyses.

The UIP group is composed of 4 patients with histological evidence of “possible UIP” or “UIP” diagnosis in the absence of lung cancer. The baseline mortality prediction model was assessed by measuring the gender age physiology (GAP) index (Table 2). The histological material taken was analyzed for diagnostic purposes, according to the ATS/ERS/JRS/ALAT guidelines [31,32].

In the UIP + LC group, composed of 11 patients, the predominant histotype was adenocarcinoma (73% of cases), while squamous cell carcinoma was less frequent (27%). The associated fibrotic histotypes were classified as “possible UIP” (64%) and “UIP” (36%). Among the patients in the UIP + LC group, tumor was classified as stage II in 64% of patients and stage I in the remaining 36%. All patients in the group UIP + LC showed an interalveolar tumor dissemination (STAS spread through air spaces), according to the staging [33].

### 4.2. Bioinformatic Analysis

Normalized gene expression profiling of healthy subjects and IPF patients were retrieved from the GEO database (https://www.ncbi.nlm.nih.gov/geo/ accessed on 1 October 2022) with accession ID GSE150910, GSE94555, and GSE124685. Log2-trasformed expression of B4GALT1 was compared between subgroups and statistical significance was assessed by the Wilcoxon rank sum test and Student’s *t*-test [34]. Mean expression of genes involved in specific pathways was correlated to B4GALT1 expression by using Pearson’s correlation. Pathways were downloaded from the Molecular Signatures Database (MSigDB v7.5.1, https://www.gsea-msigdb.org/gsea/msigdb/ accessed on 1 October 2022), and network analysis was performed using positive correlated EMT genes with B4GALT1 by using OmicsNet 2.0 and Cytoscape 3.9.1.

### 4.3. Primary Cell Cultures

Primary cell cultures were established from lung tissue obtained from two patients undergoing diagnostic thoracoscopy or surgical lobar resection of lung cancer. “Normal” lung fibroblast cell culture was obtained from the macroscopically and histologically normal part of the pulmonary lobe, surgically removed, far from neoplastic lesion. In IPF patient, the lung tissue was obtained from surgical biopsies performed for diagnostic purposes. After tissue pickup, the material underwent an enzymatic digestion with trypsin/EDTA for 1 h in 37 °C. Dissociation was stopped by adding collagenase I, a digestive inhibitor, and after centrifugation of cellular suspension, cells were plated at a concentration of 50,000 cells/mL in adherent condition with RPMI-1640 (Sigma, St. Louis, MO, USA) at FBS 10% (Gibco, Waltham, MA, USA), 1% penicillin/streptomycin (Gibco, Waltham, MA, USA), and 2 mM glutamine (Gibco, Waltham, MA, USA) in standard conditions at 37 °C 5% CO_2_. The cells were analyzed at passage number three of cultures.

### 4.4. RNA Extraction and Real Time RT-PCR

Total RNA from adherent cells was extracted using TRIzol (Thermofisher, Waltham, MA, USA) according to manufacturer’s instructions. A total of 1 µg of RNA was digested with gDNA Eraser, and first strand cDNA was synthesized with PrimeScript RT (Takara Bio, Inc. Kusatsu, Shiga, Japan). The cDNA was used for RT-PCR experiments carried out in a 7500 StepOneSystem (Applied Biosystem, Waltham, MA, USA). Histone 3 (H3) was included for the normalization of the RT-PCR. The relative amount of all mRNAs was calculated using the comparative method (2^−ΔΔCt^). Three independent experimental replicates were performed.

### 4.5. Immunofluorescence

For actin fiber visualization, cells were fixed by immersion in 4% paraformaldehyde in PBS for 30 min, rinsed with PBS and permeabilized in 0.1% Triton for 30 min. The chamber was then immersed in a solution with Bodipy FL phallacidin and rhodamine phallodin (Invitrogen, Waltham, MA, USA) for 30 min at room temperature. Images of immunofluorescence were visualized using a confocal microscope Leica (laser scanning TCS SP2) equipped with Ar/ArKr and HeNe lasers and acquired utilizing the Leica confocal software. They were observed in 20× microscope fields randomly taken from three different experiments.

### 4.6. Immunohistochemistry

Immunohistochemistry from formalin-fixed paraffin-embedded (FFPE) lung sections was performed with the following primary antibodies: cytokeratin 7 (CK7) (Agilent, Santa Clara, CA, USA, clone OV-TL 12/30), actin smooth muscle (α-SMA) (Agilent, clone 1A4), and B4GALT1 (abcam, clone ab121326). Briefly, FFPE tissue blocks were sectioned (2 m) and then submitted to deparaffinization, rehydration, and antigen retrieval by PT link (Dako Omnis-Agilent, Santa Clara, CA, USA, PT10126), according to the recommendations of the primary antibody datasheets. Endogenous peroxidase activity was quenched with methanol–hydrogen peroxide (3%) for 10 min. Tissue slides were incubated with the primary antibody in a moist chamber at 4 °C. The immune reaction was visualized by using EnVision™ FLEX/HRP (EnVision™ FLEX; Agilent, Santa Clara, CA, USA, K8000).

### 4.7. Optical Microscopy

Morphological analysis of adherent primary cell cultures was carried out by optical images by Nikon Coolpix 995, digital camera equipped with optical microscope Zeiss Axiovert. The expression of B4GALT1, CK7, and α-SMA staining intensity on immunohistochemistry was assessed on fully digitalized whole sections by analyzing the staining signal intensity using an Aperio slide scanner and subsequently exported on Aperio image scope v11 (Aperio, Leica Biosystem Techology, Wetzlar, Germany).

### 4.8. Statistical Analysis

All experiments were performed in triplicate. Results are expressed as the average +/− standard error. Statistical analysis was performed using two-tailed Student’s *t*-test. All values shown in the figures and text are *p* < 0.05, and value was assumed as statistically significant.

### 4.9. Ethics

The study was approved by the Institutional Ethical Committee of our University La Sapienza of Rome, Italy, on the basis that it complied with the declaration of Helsinki and that the protocol followed existing good clinical practice guidelines and informed consent was obtained CE 6278 and 5176/2013 from each individual.

## 5. Conclusions

In conclusion, our study was focused on the expression analysis of B4GALT1 in IPF patients. We found an increased level of mRNA and protein in lung biopsy tissues and primary cell culture obtained from IPF patients. Positive staining was detected also in lung tissue specimens from patients showing IPF in association with lung cancer. Network analysis indicated that B4GALT1 upregulation was related to genes belonging to the EMT pathway, known to be involved in both fibrosis and cancer. Although the study was conducted in a limited number of cases, the overexpression of B4GALT1 in IPF observed in our patients is a novel finding and it could represent a new diagnostic marker, potentially usable as a therapeutic target for a disease still burdened with an unfavorable prognosis. Further studies are needed to evaluate its possible use for these purposes.

## Figures and Tables

**Figure 1 ijms-23-15040-f001:**
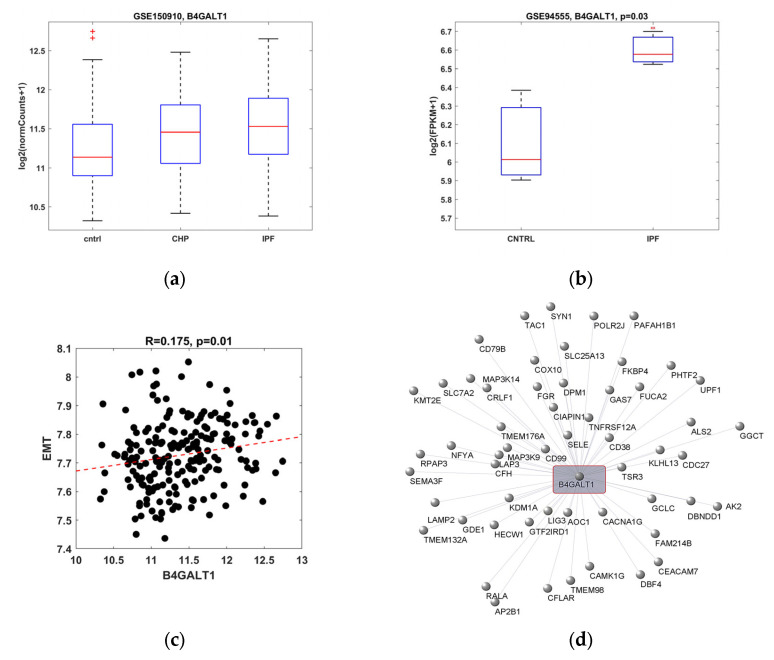
Bioinformatic analysis to evaluate B4GALT1 in silico expression. (**a**) Box plot representing B4GALT1 expression from RNA-seq-based transcriptomic analysis for whole lung tissues from 82 chronic hypersensitivity pneumonitis (CHP), 103 IPF, and 103 control subjects. We observed a statistical significance comparing both CHP and IPF samples to healthy subjects (*p* = 7 × 10^−3^, *p* = 4.1 × 10^−5^, respectively) by using the Wilcoxon rank sum test. No difference was revealed between CHP and IPF samples (*p* = 0.15). A Kruskal–Wallis test among three groups was also evaluated (*p* = 1.2 × 10^−4^). Data from the GEO database with access ID GSE150910. (**b**) Box plot representing B4GALT1 expression from scRNA-seq transcriptomic analysis of normal and IPF respiratory epithelial cells obtained from peripheral lung of controls (n = 3) and IPF patients (n = 3), respectively. Statistical significance was assessed by Student’s *t*-test. Data from GEO database with access ID GSE94555. (**c**) Scatter plot of Pearson’s correlation between B4GALT1 expression and the mean expression of HALLMARK_EPITHELIAL_MESENCHYMAL_ TRANSITION genes. Data from GEO database with access ID GSE150910. (**d**) Graphical network analysis obtained by Pearson’s correlation between B4GALT1 and 55 positive correlated genes involved in the EMT pathway.

**Figure 2 ijms-23-15040-f002:**
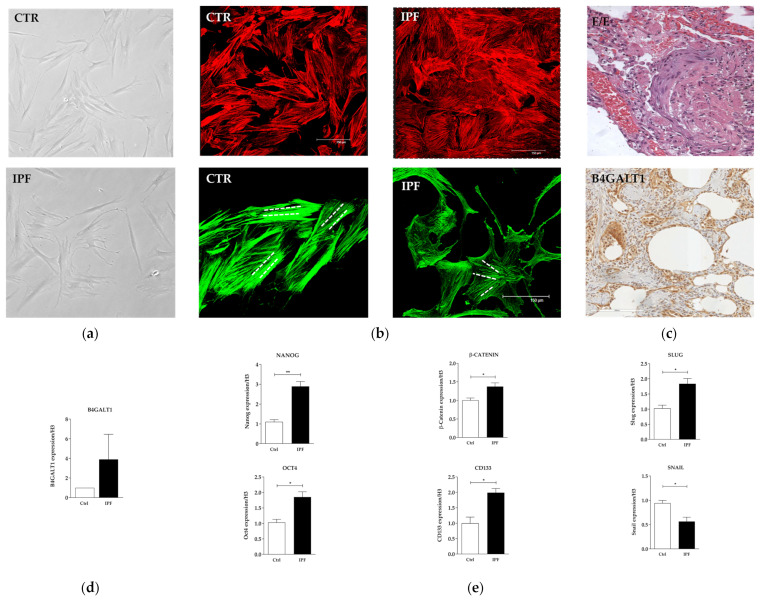
In Vitro characterization of IPF primary cell line. (**a**) Primary culture of human normal fibroblast (upper) and IPF (lower) (×10 magnification). (**b**) Representative bright field images on actin of normal fibroblast and IPF primary cell cultures (×40 magnification). (**c**) H&E corresponding to lung biopsy of the patient with UIP from which the primary cell culture was isolated and IHC analysis (×20 magnification). (**d**) mRNA expression level of B4GALT1. (**e**) mRNA levels of stemness genes and EMT genes between normal fibroblast and IPF verified by qRT-PCR. The data are expressed as mean ± SD.* *p* < 0.05; ** *p* < 0.005.

**Figure 3 ijms-23-15040-f003:**
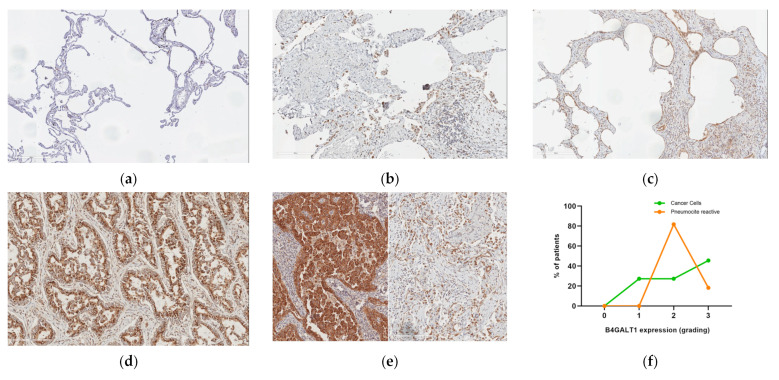
B4GALT1 expression in lung fibrosis and lung cancer (×20 magnification). (**a**) An example of near-normal lung tissue is shown (B4GALT1 staining); (**b**) positive staining in lung cancer; (**c**) tumor (on the left) and UIP (on the right) of the same patient show simultaneous strong cytoplasm B4GALT1 expression; (**d**) strong B4GALT1 expression is observed in most fibroblastic foci in UIP both in connective tissue and alveolar epithelium; (**e**) and possibly in UIP fibrous reactive remodeling of lung tissue not related to IPF; (**f**) quantification of grading expression of B4GALT1; (**g**) there are B4GALT1-positive immune lymphocyte infiltrating cells in neoplastic and UIP tissue samples; and (**h**,**i**) capillary endothelium expressing B4GALT1 was observed, in the same patient, both in the immediately peritumoral connective tissue area and adjacent to the UIP pattern fibrous.

**Table 1 ijms-23-15040-t001:** Common genes and pathways in correlation between B4GALT1 and EMT.

Pathways	*p*-Value	q-Value	Genes
Focal adhesion (WP306)	3.54 × 10^−28^	1.04 × 10^−25^	ITGB1, ITGB5, LAMA2, LAMA1, ITGB3, LAMA3, TNC, LAMC2, LAMC1, THBS2, THBS1, MYLK, COMP, SPP1, FLNA, ITGAV, PDGFRB, JUN, ITGA2, FN1, VEGFC, VEGFA, COL1A1, COL1A2, COL4A2, COL4A1, COL5A3, COL6A2, COL5A2, ITGA5, MYL9
miRNA targets in ECM and membrane receptors (WP2911)	1.87 × 10^−23^	2.77 × 10^−21^	ITGB5, FN1, LAMC1, THBS2, THBS1, COL3A1, COL1A2, COL5A1, COL4A2, COL4A1, COL5A3, COL6A2, COL5A2, COL6A3
Focal adhesion-PI3K-Akt-mTOR-signaling pathway (WP3932)	1.93 × 10^−22^	1.89 × 10^−20^	ITGB1, ITGB5, LAMA2, LAMA1, COL11A1, ITGB3, LAMA3, TNC, LAMC2, LAMC1, THBS2, FGF2, THBS1, COMP, SPP1, ITGAV, PDGFRB, ITGA2, FN1, VEGFC, VEGFA, COL1A1, COL3A1, COL1A2, COL4A2, COL5A1, COL4A1, COL5A3, COL6A2, COL5A2, ITGA5
PI3K-Akt signaling pathway (WP4172)	7.67 × 10^−19^	5.65 × 10^−17^	ITGB1, ITGB5, LAMA2, LAMA1, ITGB3, LAMA3, TNC, LAMC2, LAMC1, THBS2, FGF2, THBS1, COMP, SPP1, ITGAV, PDGFRB, BDNF, ITGA2, FN1, VEGFC, VEGFA, COL1A1, IL6, COL1A2, COL4A2, COL4A1, COL6A2, COL6A3, ITGA5
IL-18 signaling pathway (WP4754)	4.42 × 10^−16^	2.60 × 10^−14^	JUN, CXCL8, SDC4, MMP1, MMP2, MMP3, FN1, PLOD3, TNFAIP3, TNFRSF11B, VEGFA, COL1A1, ACTA2, MMP14, COL3A1, IL6, COL1A2, SPP1, TIMP3, FAS, PTX3, TIMP1, SNTB1, TGM2
Type I collagen synthesis in the context of osteogenesis imperfecta (WP4786)	1.21 × 10^−14^	5.95 × 10^−13^	COL1A1, COL1A2, BMP1, LOX, SERPINH1, P3H1, PLOD2, TNFRSF11B, PLOD1, PPIB, COLGALT1
Senescence and autophagy in cancer (WP615)	1.48 × 10^−13^	6.27 × 10^−12^	JUN, SPARC, TGFB1, CXCL8, IGFBP3, SERPINE1, FN1, CXCL1, INHBA, THBS1, COL1A1, IL6, MMP14, COL3A1, CD44
Arrhythmogenic right ventricular cardiomyopathy (WP2118)	9.15 × 10^−12^	3.37 × 10^−10^	ITGB1, GJA1, SGCD, ITGB5, CDH2, LAMA2, SGCB, ITGB3, ITGA2, ITGAV, ITGA5, SGCG
Small cell lung cancer (WP4658)	1.31 × 10^−11^	4.32 × 10^−10^	ITGB1, LAMA2, GADD45B, LAMA1, GADD45A, ITGA2, LAMA3, FN1, LAMC2, LAMC1, COL4A2, COL4A1, ITGAV
Lung fibrosis (WP3624)	6.43 × 10^−10^	1.48 × 10^−8^	GREM1, IL6, CXCL8, TGFB1, ELN, MMP2, SPP1, PTX3, TIMP1, FGF2

Enrichment analysis by WikiPathways2021 Human by Appyters Software [6].

**Table 2 ijms-23-15040-t002:** Characteristics of UIP/IPF and UIP/IPF cancer -associated patients recruited for the study.

**Information**					**Histology**		**Spirometry**		**Radiology**					
**ID**	**Age**	**Sex**	**Smoke**	**Pack Year**	**Diagnosis**	**Gap Index and Stadiation**	**FEV1 (%)**	**FVC (%)**	**Ct Score**	**Ground Glass**	**Honey Combing**	**Consolidations**	**Cross Linking**	**Enphysema**
UIP1	72	M	-	-	Possible UIP	II	90.5	77.8	30	No	No	No	Yes	No
UIP2	72	M	Yes	37.5	UIP	I	110	98	45	No	Yes	No	Yes	No
UIP3	78	F	Yes	3.75	Possible UIP	I	113.9	122.30	20	No	Yes	No	Yes	No
UIP4	49	M	Yes	15	UIP	I	56	54	50	Yes	Yes	No	Yes	No
UIP + LC1	75	M	-	40	Lung AdK + UIP	IA2	129.30	154.70	15	No	No	Yes	Yes	Yes
UIP + LC2	64	M	Yes	30	Lung squamous K + possible UIP	IB	71	68	80	Yes	Yes	No	Yes	Yes
UIP + LC3	76	M	-	60	Lung AdK + possible UIP	IB	76.7	70.4	40	Yes	Yes	Yes	Yes	Yes
UIP + LC4	63	M	Yes	34	Lung squamous K+ possible UIP	IIB	77.60	79.60	30	No	Yes	No	Yes	No
UIP + LC5	74	M	-	30	Lung AdK + UIP	IIB	92	85	80	Yes	Yes	Yes	Yes	No
UIP + LC6	68	M	-	5	Lung AdK+ UIP	IB	108	100	20	Yes	No	Yes	Yes	No
UIP + LC7	79	M	Yes	20	Lung AdK + UIP	IA2	129.1	119.1	15	Yes	Yes	Yes	Yes	No
UIP + LC8	80	M	-	-	Lung squamous K+ Possible UIP	IB	88.4	81.1	-	-	-	-	-	-
UIP + LC9	81	M	No	-	Lung AdK + possible UIP	IA2	100.6	89.9	5	Yes	No	Yes	No	No
UIP + LC10	67	M	-	80	Lung AdK + possible UIP	IA3	70	112	20	No	No	Yes	Yes	Yes
UIP + LC11	78	M	-	-	Lung squamous K + possible UIP	IIB	63.10	81.20	50	No	Yes	Yes	Yes	Yes

Data of the main epidemiological characteristics, smoking habit (measured in p/Y), histological diagnosis, respiratory function tests, as well as radiological investigations at the time of biopsy of enrolled IPF patients. Patients were divided into the UIP and UIP + lung cancer (LC) groups.

## Data Availability

The datasets used and/or analyzed during the current study are available from the corresponding author on reasonable request. All data generated or analyzed during this study are included in this published article.

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
