# Peer review of "B4GALT1 as a New Biomarker of Idiopathic Pulmonary Fibrosis"

_ijms, 2022, doi:10.3390/ijms232315040_

Round 1
Reviewer 1 Report
The papers entitled “B4GALT1 as a new biomarker of idiopathic pulmonary fibrosis” demonstrated that B4GALT1 is overexpressed at mRNA and protein levels in lung specimens from IPF patients. Authors concluded that the expression of B4GALT1 could represent a novel diagnostic marker for IPF.
I have a few questions below.
1. Not sure if B4GALT1 has a more significant impact in lung cancer?
2. Is there a significant correlation between a lung cancer patient's survival and his B4GALT1 expression?
3. What is the prognosis of the 11 lung cancer patients in this study?
4. This study showed UIP-specific B4GALT1 expression. What about other expressions in her IIP like NSIP?
Author Response
- The clinical impact of B4GALT1 in lung cancer still needs to be elucidated in appropriate prospective prognostic studies. The relevance of the expression of this gene in lung cancer is mainly based on retrospective IHC studies performed by our group and others. In addition, its role was assessed in studies performed on dataset repository (A Web Server for Cancer and Normal Gene Expression Profiling and Interactive Analyses. Nucleic Acids Res. 2017, 45, W98–W102.Tang, Z.; Li, C.; Kang, B.; Gao, G.; Li, C.; Zhang, Z. GEPIA: A Web Server for Cancer and Normal Gene).
- We tried to correlate B4GATL1 expression with lung cancer patient’s survival and we did find a correlation that didn’t rich significant because of insufficient number of patients considered.
- We didn’t report the correlation between B4GALT1 and prognosis in our group of patients with lung cancer associated with IPF because we couldn’t provide the outcome in all these patients. In three of them in fact, we had no information regarding mortality or other information on the clinical condition of the patients. In the remaining eight patients we observed mortality, in two of them. Since the data were not completed we didn’t mention in the manuscript such results and we think that should be more appropriate to perform a dedicated analysis to answer this question.
- We didn’t analyzed the expression of B4GALT1 in other IIP lesions such as NSIP. However, we noted B4GALT1 expression, at variable degree and to a lesser extent, in fibrotic areas associated with lung cancer, implying active remodelling of lung parenchyma.

Reviewer 2 Report
The authors present an interesting and well written study about B4GALT1 as a new biomarker of idiopathic pulmonary fibrosis.
- The authors should describe the primary fibroblast culturing in more detail. Where the cell obtained from parenchyma or more central areas of the lung. In which passage(s) was the cells used for the analyses, cell concentration,
- Protein and gene expression analysis was also performed in patients with fibrosis in other non-lung tissues. The authors should specify from how many patients (n=? in the result section) in each organ. The authors only present images from the liver from two cases in S1. Additional representative images should be presented from the other organs in the supplement. Did the authors also observe alterations in the kidneys?
- The reference list is not totally aligning to the Vancouver style. Ref 9 is missing journal and pages. The authors should update the reference list
- There are some minor spelling that needs to be corrected.
Author Response
1- The cells were obtained from peripheral areas of the lung, and analyzed early after few passages of culture at a concentration of 50000 cells/ml. we stated in the material and methods section.
2- Analysis of B4GALT1 expression was performed in a total of three cases of each different condition considered. In the supplementary figure, we reported the most representative figure regarding two cases fibrosis of the kidney. We changed the text by including the number of the cases as suggested by the reviewer.
3- The reference list was corrected according to the Vancouver style guidelines, and in additional reference was added at position 30. All the reference number was modified accordingly in reference list as well as in the text.
4- We corrected minor spelling errors in the text.

Round 2
Reviewer 2 Report
The authors have made some changes.
The authors should highlight and add to the conclusion that this is a limited study with few observations/limited patient material.
Line 141: To confirm the immunophenotype pattern of IPF, two specific markers where evaluated by immunohistochemistry (IHC). Both of them, namely α-SMA and CK7, where higly expressed (data not shown). The authors should show these stainings in the supplement. An h is also missing in highly.
The authors should specify in more detail what they mean with a few passages for the fibroblast cell cultures as the fibroblasts loose their phentype over time. Do the authors refer to passage 2-3?
Author Response
<
According to the reviewer suggestion, we included in the conclusion section a sentence indicating that the study was conducted on a limited number of cases.
We corrected the misspelling reported by the reviewer, we included a new supplementary figure 1 showing our IHC analysis of α-SMA and CK7 performed on our lung tissues. The number of supplementary figure as been modified accordingly in the text.
We specified in the text the number of passages of our fibroblast cultures
According to the reviewer suggestion, we included in the conclusion section a sentence indicating that the study was conducted on a limited number of cases.
We corrected the misspelling reported by the reviewer, we included a new supplementary figure 1 showing our IHC analysis of α-SMA and CK7 performed on our lung tissues. The number of supplementary figure as been modified accordingly in the text.
We specified in the text the number of passages of our fibroblast cultures.